# *Candida auris* Dry Surface Biofilm (DSB) for Disinfectant Efficacy Testing

**DOI:** 10.3390/ma12010018

**Published:** 2018-12-21

**Authors:** Katarzyna Ledwoch, Jean-Yves Maillard

**Affiliations:** School of Pharmacy and Pharmaceutical Sciences, Cardiff University, Cardiff CF10 3NB, UK; LedwochK@cardiff.ac.uk

**Keywords:** *Candida auris*, dry-biofilm, disinfection, peracetic acid, sodium hypochlorite, chlorine dioxide, sodium dichloroisocyanurate, transferability, regrowth

## Abstract

*Candida auris* is an emerging pathogen that needs to be controlled effectively due to its association with a high mortality rate. The presence of biofilms on dry surfaces has been shown to be widespread in healthcare settings. We produced a *C. auris* dry surface biofilm (DSB) on stainless steel surfaces following sequential hydration and desiccation cycles for 12 days. The ASTM2967-15 was used to measure the reduction in viability of 12 commercially wipe-based disinfectants and sodium hypochlorite (1000 ppm) against *C. auris* DSB. We also evaluated *C. auris* transferability and biofilm regrowth post-treatment. A peracetic acid (3500 ppm) product and two chlorine-based products (1000 ppm available chlorine) were successful in reducing *C. auris* viability and delaying DSB regrowth. However, 50% of the products tested failed to decrease *C. auris* viability, 58% failed to prevent its transferability, and 75% did not delay biofilm regrowth. Using three different parameters to measure product efficacy provided a practical evaluation of product effectiveness against *C. auris* DSB. Although log_10_ reduction in viability is traditionally measured, transferability is an important factor to consider from an infection control and prevention point of view as it allows for determination of whether the surface is safe to touch by patients and hospital staff post-treatment.

## 1. Introduction

*Candida auris* was first isolated and identified in Japan in 2009 [1]. *C. auris* is an emerging pathogen responsible for many life-threating infections and it can be associated with high mortality rates [2]. *C. auris* infections are difficult to treat mostly due to the unpredictable resistance profile of the yeast to anti-fungal agents, frequent misidentification, non-aggregative phenotype, and its ability to form biofilm [3]. Higher risk of candidemia occurs in immunocompromised patients, patients that have undergone antibiotic or anti-fungal therapy, patients after surgeries, and patients with central venous catheters [4].

Contaminated surfaces in healthcare settings contribute to the transmission of infectious diseases [5,6,7]. Vickery and colleagues [8] showed that pathogens embedded in dry surface biofilms (DSB) remain on hospital surfaces despite rigorous surface decontamination. The widespread presence of dry surface biofilms on healthcare surfaces has now been established [9,10]. *C. auris* can persist on surfaces for weeks [11,12] and transmission of *C. auris* in healthcare settings have been reported [2,4,13,14]. The elimination of *C. auris* from surfaces is therefore important to consider. However, not many studies have investigated the effectiveness of disinfectants against *C. auris* [15]. Public Health England guidance [16] for the management and infection prevention and control of *C. auris* recommends using hypochlorite at 1000 ppm available chlorine to terminally clean room or bed space after the discharge of a *C. auris* infected or colonized patient. Not surprisingly, the majority of efficacy studies against *C. auris* relates to chlorine-releasing agents [14,17,18,19], although other biocides have been considered, such as quaternary ammonium compounds [14,18], acetic acid [18], peracetic acid [19], and hydrogen peroxide [17]. All these studies but one studied planktonic (suspension of) *C. auris* [20]. None of these studies investigated the transferability of *C. auris* to other surfaces post-treatment. Here, we investigated the efficacy of 12 commercially available products and sodium hypochlorite 1000 ppm against *C. auris* DSB using a modified product efficacy test protocol ASTM2967-15 [21] to measure decreases in viability, transferability, and biofilm regrowth post-treatment. The evaluation of three different parameters provides a better and more practical understanding of product efficacy against this pathogen.

## 2. Materials and Methods

### C. auris Growth and Maintenance

*C. auris* (DSM 21092) was propagated overnight in malt extract broth (MEB, Oxoid, Thermo Scientific™, Loughborough, UK) at 25 °C and the pellet was re-suspended in MEB following centrifugation at 1200× *g*. The yeast suspension was adjusted to 1 × 10^6^ CFU/mL.

### C. auris Organic Load (OL) Dry-Biofilm Model

The *C. auris* DSB model is based on a recently developed *Staphylococcus aureus* DSB protocol [22]. Briefly, dry-biofilm formation consists of alternating hydration and desiccation phases in the presence of an organic load (OL). Stainless steel AISI 430 discs (0.7 ± 0.07 mm thickness; 10 ± 0.5 mm diameter, Goodfellow Cambridge Limited, Huntington, UK) were used as support. Sterile discs were placed in wells of a Corning™ Costar™ flat-bottom cell culture plates (Fisher Scientific™, Loughborough, UK), containing 1 mL of MEB with 5% anhydrous D-glucose (Fisher Scientific, Loughborough, UK), 3 g/L bovine serum albumin (BSA; Sigma^®^ Life Science, Dorset, UK), and 10^6^ CFU/mL washed *C. auris* suspension. Yeasts were first allowed to attach and form a biofilm on the disc surface for 2 days at 25 °C under gentle agitation using an Orbit P4 plate rocker (Labnet International, Edison, NJ, USA). The suspension was then drained from the wells, and plates were incubated at 25 °C for 48 h. Following this dry phase, 1 mL of MEB with 3 g/L BSA was added to each well, and a new hydrated phase began for 48 h. Hydrated and dry phases alternated every 48 h for a period of 12 days, ending with biofilm in a dry phase.

### Scanning Electron Microscopy (SEM) Imaging

*C. auris* DSB samples were prepared by overnight incubation of discs in a 2.5% glutaraldehyde solution (ACROS Organics™, Fisher Scientific, Loughborough, UK) followed by immersion in successive concentrations of 10%, 25%, 50%, 70%, 90%, and 100% ethanol (Honeywell, Fisher Scientific Ltd., Loughborough, UK) for 10 min each. Prior to scanning electron microscopy (SEM, Carl Zeiss Ltd., Cambridge, UK) scanning, samples were coated with 20 nm of AuPd coating with a sputter coater (SC500, Biorad, UK). Secondary electron images were acquired with a beam energy of 5 kV using an in-lens detector on a Sigma HD Field Emission Gun Scanning Electron Microscope (Carl Zeiss Ltd., Cambridge, UK) at ×2000 and ×10,000 magnification and a 5 mm working distance. SEM images were false-colored to help visualization and contrast using GNU Image manipulation program (GIMP 2.8) software. Images were not otherwise altered. 

### Product Tested

The effectiveness of four commercially available wipes and eight commercially available liquid disinfectants was tested against *C. auris* OL dry-biofilm (Table 1). Disinfectants were prepared according to manufacturers’ instructions and combined with Rubbermaid^®^ HYGEN™ disposable microfiber cloth (Rubbermaid Products, Surrey, UK), allowing 2.5 mL of disinfectant per 1 g of wipe. Wipes were cut into 3 × 3 cm^2^ squares prior to testing. 

### ASTM E2967-15 Test

Disinfection tests were performed according to a modified ASTM E2967-15 test [21]. Briefly, DSB were wiped with the Wiperator (Filtaflex Ltd., Almonte, Ontario, Canada) from both sides for 10 s under 500 g pressure, left for 2 min at 25 °C, and then neutralized in Dey-Engley neutralizing (DE) broth (Neogen^®^ Corporation, Ayr, UK). Transfer of viable yeasts from used wipes to clean sterile disc was not performed.

### Reduction in Viability for Yeasts Embedded in Dry Biofilms

Following wiping, samples were incubated for 1 h at 25 °C in 2 mL DE with 100 µg/mL proteinase K (Fisher Bioreagents™, Fisher Scientific, Loughborough, UK) and 1 g of glass beads (Fisher Scientific, Loughborough, UK). After incubation, samples were vortexed for 2 min and then serially diluted, and 3 × 10 µL^2^ drops of each dilution was plated onto tryptone soya agar (TSA; Oxoid, Thermo Fisher Scientific, Newport, UK). Reduction in yeast viability, expressed as a log_10_ reduction, was calculated as the difference between the number of yeasts recovered from untreated (control) and treated samples.

### Transferability Test

Following wiping, discs were pressed 36 separate times with 100 g pressure on the surface of DE agar. Following the transfer test, DE agar was incubated at 25 °C for a up to 5 days until colonies appeared. Positive growth/adpression was recorded, and transferability was calculated as the number of positive contact/number of adpressions. 

### Dry-Biofilm Regrowth

Following wiping, discs were placed in 30 mL capacity flat bottom glass bottle with 2 mL of DE broth. The number of days for turbidity change, which is indicative of growth, was recorded. Samples were plated on TSA to confirm yeast growth and purity.

### Statistical Analysis

The statistical significance of data sets was evaluated with GraphPad PRISM^®^ (version 7.04, GraphPad Software, San Diego, CA, USA) using two- and one-way Analysis of Variance (ANOVA). All experiments were performed in triplicates in three independent biological replicates unless otherwise stated. The sample standard deviation was evaluated with Bassel’s correction. 

## 3. Results

### SEM Analysis of C. auris Dry Surface Biofilm

*C. auris* formed a thin biofilm that was evenly scattered throughout the stainless-steel disc surface with no evidence of extracellular polymeric substances (Figure 1). There was no statistically significant difference (two-way ANOVA, *p* = 0.06) in viable count of yeasts (log_10_ CFU/mL = 7.8 ± 0.3) recovered from each disc between four independent biofilm batches.

### Product Efficacy

The most effective treatments including PAA-1 (3500 ppm), NaDCC-5 (1000 ppm), NaOCl-Ref (1000 ppm), and NaOCl-3 (1000 ppm) removed or killed more than 7 log_10_ of *C. auris* embedded in DSB (Figure 2). Peracetic acid at 3500 ppm combined with a non-woven wipe was significantly (one-way ANOVA, *p* < 0.05) more effective in biofilm eradication than PAA at 250 ppm combined with a microfiber cloth (0.84 ± 0.11 log_10_ reduction). NaDCC-5 was the most effective (two-way ANOVA, *p* < 0.05) in killing or removing *C. auris* DSB compared to the other NaDCC-based products that all failed to produce a 4 log_10_ reduction in viability (Figure 2). Chlorine-dioxide-based products overall did not perform very well, achieving less than 2.5 log_10_ reduction even with ClO_2_-2 containing a higher concentration (1000 ppm) of available chlorine (Figure 2). There was no difference (one-way ANOVA, *p* = 0.22) in activity between ClO_2_-1 and ClO_2_-2. Overall, half of the products tested (ClO_2_-1, NaDCC-2, and NaOCl-2) showed either a similar performance (one-way ANOVA, *p* > 0.05) to water combined with the microfiber cloth or performed worse (one-way ANOVA, *p* < 0.05) than water combined with the microfiber cloth (ClO_2_-2, NaDCC-3, and PAA-2).

Only two products, PAA-1 and NaOCl-3, prevented *C. auris* transfer after treatment (Figure 3). Seven out of 12 commercially available disinfectants were not effective in lowering the transferability of *C. auris* from DSB post-wiping. For four product/materials combinations (ClO_2_-2, NaDCC-2, NaDCC-3, and NaOCl-1), there was no statistically significant difference (one-way ANOVA, for each pair *p* > 0.05) between their performance and that of water (Figure 3). The remaining three treatments (NaDCC-4, NaOCl-2 and PAA-2) were even less effective than wiping with water (one-way ANOVA, for each pair *p* < 0.05).

The best commercial products, NaDCC-3, NaDCC-5, and PAA-1, delayed the recovery of biofilm post-treatment for more than 4 days (5.0 ± 0.0, 4.7 ± 1.2 and 6.5 ± 2.1 days, respectively; Figure 4). Such activity was similar (two-way ANOVA, *p* = 0.53) to NaOCl-Ref which delayed regrowth by 4.5 ± 0.7 days. Nine out of 12 commercial treatments failed to prevent the regrowth of *C. auris* DSB for more than 2 days (Figure 4).

The less-reactive chemistry (as compared to the oxidizing chemistries) based on a quaternary ammonium compound produced a 4 log_10_ reduction in *C. auris* on surfaces and reduced transferability post-wiping to 20%.

## 4. Discussion

Environmental surfaces play an important role in the transmission of infection [5,6]. Infection control regimens include the combination of cleaning and disinfection processes [16] and are often based on the use of a disinfectant or cleaner combined with diverse materials [31]. Performance of biocidal products still relies, however, on testing the efficacy of the formulation and not the combination of formulation and material [22]. A number of US-based protocols to test wipe activity have been described, but all had severe limitations in their setting or performance [31]. Recently, the ASTM2967-15 and the EN16615-15 protocols [32] have been recommended for evaluating the efficacy of antimicrobial wipes, although EN16615-15 has recently been shown to lack stringency [33]. Both tests evaluate the reduction in microbial number from the test surface and the transfer of micro-organisms during wiping. Attaining a > 5log_10_ reduction in microbial inoculum following treatment has been, until now, deemed to provide enough assurance that all micro-organisms would be killed on surfaces in practice. Although this might be the case where surfaces are contaminated with a low number of microorganisms, this might not be correct with surfaces contaminated with a high number of microorganisms or when dry surface biofilms are present. Hence, the evaluation of microbial transferability post-treatment is important to take into consideration to provide reassurance that a surface would be safe to touch. The presence of DSB on healthcare surfaces has been established [8,9,10], and the resilience of DSB to disinfection has been described [8]. The presence of *C. auris* DSB has not been yet established in healthcare settings, although its persistence in healthcare settings has been described [11,12].

Here, we have successfully produced reproducible dry surface biofilms of *C. auris* on stainless steel surfaces, for which appearance and characteristics are not dissimilar to artificial DSB of *Staphylococcus aureus* (Ledwoch, Said and Maillard, unpublished results) or DSB isolated from endoscopes [34]. These dry surface biofilms of *C. auris* provided a platform for testing the efficacy of commercially available wipe-based products or formulation combined with a microfiber cloth. Here, we observed that the majority of commercially available chlorine-releasing agents widely used by hospitals do not effectively eradicate a *C. auris* DSB or lower its transferability. The ability of *C. auris* to form dry surface biofilm on surfaces contribute somewhat to its resistance against disinfection. We showed that *C. auris* inactivation was product dependent. NaOCl-containing products are widely used in healthcare settings, although the efficacy of NaOCl against *C. auris* dried on surfaces differs in the literature. While only 2.5–3 log_10_ of *C. auris* were killed by NaOCl (1000 ppm) within 5 min [20], a 3000 ppm NaOCl solution resulted in reducing *C. auris* dried on surfaces for 2 h by 6 log_10_ in 1 min [18]. An 8000 ppm NaOCl solution, however, produced a 6 log_10_ reduction in 10 min for *C. auris* dried on surfaces for 1 h [14]. Here, NaOCl-Ref (1000 ppm) and NaOCl-3 (1000 ppm) produced a 7 log_10_ reduction in number within 2 min following wiping. 

The microfiber cloth loaded with sterile water enabled the removal of *C. auris* (2 log_10_ reduction) but failed to prevent transfer or biofilm re-growth post wiping. Interestingly, a number of products performed similarly to water, indicating either that the combination with the Rubbermaid^®^ HYGEN™ disposable microfiber cloth was incompatible with the product formulation, or that the product activity was caused by the material only. Wesgate et al. [33] recently showed that the type of material might have a small impact on the formulation presumably because the concentration of the active ingredient was high. In this study, it was interesting to observe that lower concentration and, possibly, the type of material affected the efficacy of peracetic acid and NaOCl. It has also been reported that the formulation itself impacts activity [35]. Here, different NaOCl formulations produced different results. While the use of unformulated NaOCl (NaOCl-Ref) was effective in reducing counts of *C. auris* on surfaces and prolonged biofilm regrowth, it failed to prevent *C. auris* transfer. The formulated NaOCl product (NaOCl-3) containing the same concentration of available chlorine produced a high reduction in count, thus preventing transfer but was less efficacious in delaying biofilm regrowth. NaOCl-2 containing a lower concentration of available chlorine was not effective in reducing *C. auris* count, preventing transfer, or delaying regrowth. The impact of formulation is also evident with NaDCC-based products. Overall, the peracetic acid wipe product containing 3500 ppm of PAA performed the best against *C. auris* DSB.

## 5. Conclusions

We successfully developed a dry surface biofilm in vitro model of *C. auris*. Although *C. auris* has not yet been isolated from environmental DSB, its presence on surfaces and associated high pathogenicity highlights the need to select an efficient infection control regimen. The use of a product efficacy test such as ASTM2967-15 is essential to evaluate the efficacy of formulated products. The additional evaluation of transferability of microorganisms post-wiping provides important information on a product’s overall efficacy as well as reassurance that surfaces are safe to touch post-treatment. Here, we observed that measuring log_10_ reduction in viability was not enough to discriminate between product efficacy. Importantly, we observed that a number of commercially-available formulations combined with a microfiber cloth, or products, failed to control dry surface biofilms of *C. auris*. It was also clear that high concentration and an appropriate formulation of the active ingredient was key for efficacy, with the PAA-based product performing better.

## Figures and Tables

**Figure 1 materials-12-00018-f001:**
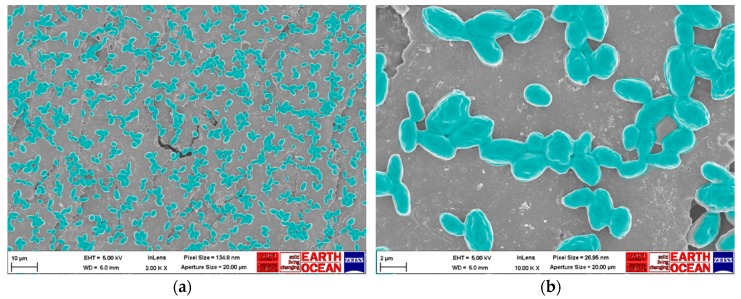
Scanning electron microscope images of *C. auris* organic load (OL) dry surface biofilm: (**a**) ×2000 magnification; (**b**) ×10,000 magnification. The images presented are representative for the whole disc surface. Observations were made on three independent triplicates of *C. auris* dry-biofilm, and whole disc surface (~0.8 cm^2^) was investigated each time with ×500 magnification. Images of dry surface biofilm (DSB) were colored in green to help visualization and contrast using GNU Image manipulation program (GIMP 2.8) software. Images were not otherwise altered.

**Figure 2 materials-12-00018-f002:**
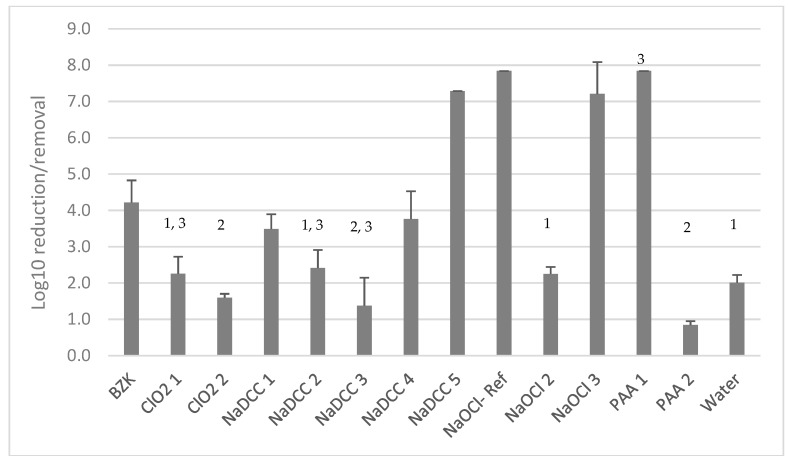
Product efficacy in killing/removing *C. auris* embedded in a DSB. 1: indicates no statistical difference (one-way ANOVA, *p* > 0.05) in log_10_ reduction/removal from surfaces; 2: log_10_ reduction/removal lower than wiping with water (one-way ANOVA, *p* < 0.05); and 3: indicates that only two biological replicates were performed.

**Figure 3 materials-12-00018-f003:**
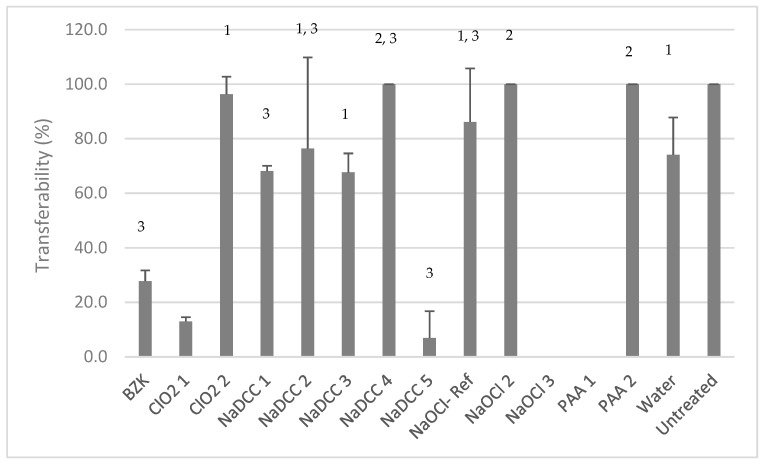
Product efficacy in preventing *C. auris* transferability post-wiping. 1: indicates no statistical difference (one-way ANOVA, for each pair *p* > 0.05) in transferability; 2: higher transferability (one-way ANOVA, for each pair *p* < 0.05) than with water control; and 3: indicates that only two biological replicates were performed.

**Figure 4 materials-12-00018-f004:**
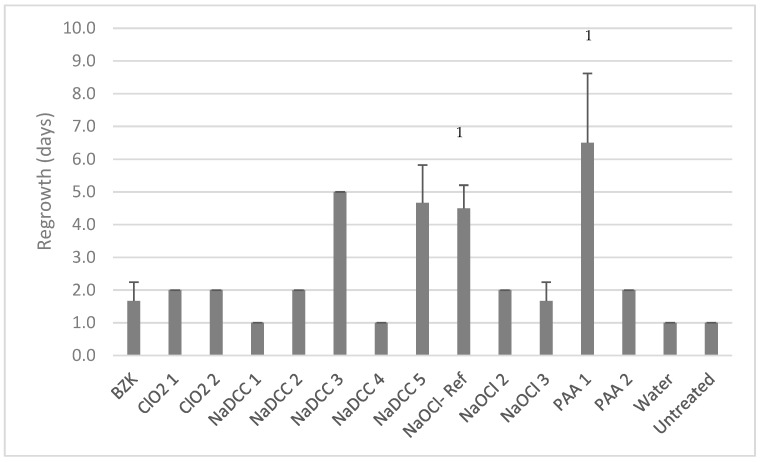
Efficacy of products in preventing regrowth post-wiping. 1: indicates that only two biological replicates were performed.

**Table 1 materials-12-00018-t001:** Disinfectants tested.

Abbreviation	Main Active Ingredient ^1^	Excipients (from MSDS) ^1^	Concentration of the Main Active Ingredient ^4^	pH ^5^	Mechanism of Disinfectant Action ^6^	Wipe Material
BZK	Benzalkonium chloride, polyhexamethylene biguanide (PHMB)	Didecyl dimethyl ammonium chloride	< 0.5% (<5000 ppm)	5.41	Membrane active agents; damage cytoplasmic membrane and increase permeability [23]	Non-Woven Wipe ^7^
ClO_2_-1	Chlorine dioxide	Sodium chlorite, sodium dodecyl sulphate, sodium carbonate, citric acid, sodium dichloroisocyanurate	300 ppm	5.05	Affect membrane permeability of the membrane and inhibits cellular respiration [23]	Microfiber cloth ^8^
ClO_2_-2	Chlorine dioxide	Not mentioned	1000 ppm	4.31		Microfiber cloth ^8^
NaDCC-1	Sodium dichloroisocyanurate	Adipic acid, arylsulfonates, sodium fatty acid sarcosides	1000 ppm	6.31	Permeabilization of the cytoplasmic membrane [24], progressive oxidation of thiol groups to disulphides [25] and deleterious effects on DNA synthesis [26]	Microfiber cloth ^8^
NaDCC-2	Sodium dichloroisocyanurate	Adipic acid, sodium toluene sulphonate, sodium n-lauroylsarcosinate	1000 ppm	5.93		Microfiber cloth ^8^
NaDCC-3	Sodium dichloroisocyanurate	Sulfonic acid	10,000 ppm	5.77		Non-woven wipe ^9^
NaDCC-4	Sodium dichloroisocyanurate	Adipic acid, sodium carbonate	1000 ppm	5.86		Microfiber cloth ^8^
NaDCC-5	Sodium dichloroisocyanurate	Adipic acid, sodium toluenesulphonate, sodium *N*-lauroyl sarcosinate	1000 ppm	5.64		Microfiber cloth ^8^
NaOCl-Ref ^2^	Sodium hypochlorite	N/A	1000 ppm	11.31	Biosynthetic alterations in cellular metabolism [27], phospholipid degradation, irreversible enzymatic inactivation in bacteria, lipid and fatty acid degradation [28]	Microfiber cloth ^8^
NaOCl-2	Sodium hypochlorite	Sodium hydroxide, sodium chloride	500 ppm	8.68		Non-woven wipe ^7^
NaOCl-3	Sodium hypochlorite	phosphoric acid (trisodium salt, dodecahydrate), sodium hydroxide, phosphoric acid	1000 ppm	13.13		Non-woven wipe ^7^
PAA-1	Peracetic acid	sodium percarbonate, citric acid	3500 ppm	8.82	Rupture or dislocation of cell wall, disruption of biochemical processes intercellularly [29] and impairment of DNA replication [30]	Non-woven wipe ^9^
PAA-2	Peracetic acid	Not mentioned	250 ppm	7.74		Microfibre cloth ^8^
Water ^3^	N/A	N/A	N/A	6.99	N/A	Microfibre cloth ^8^

^1^: Main active ingredient and excipients mentioned in the MSDS information of the commercial products used in this study. ^2^: Unformulated sodium hypochlorite (1000 ppm), used as reference. ^3^: Sterile deionized water. ^4^: Concentration of available chlorine/peracetic acid concentration was measured with Pocket Colorimeter™ (HACH^®^, Manchester, UK) (regardless of the product claim on label) via the *N*, *N*-diethyl-p-phenylenediamine (DPD) method. ^5^: pH was measured by bench top pH meter (HANNA^®^ Instruments, Leighton Buzzard, UK). ^6^: Reported mechanisms of action, mainly from studies of bacteria. ^7^: Wipe originally moisturized with disinfectant by the manufacturer. ^8^: Disinfectant prepared according to manufacturer’s instruction and then placed on Rubbermaid^®^ HYGEN™ disposable microfiber cloth (2.5 mL of liquid per 1 g of cloth). ^9^: Dry non-woven wipe impregnated with powder particles—needs to be wetted according to manufacturer instructions prior to use.

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
