# Peer review of "Candida auris Dry Surface Biofilm (DSB) for Disinfectant Efficacy Testing"

_materials, 2018, doi:10.3390/ma12010018_

Round 1
Reviewer 1 Report
The authors studied the activity of disinfectants against dry-biofilm of C. auris. They also evaluated C. auris transferability and biofilm regrowth post-treatment.
the work is interesting because of practical use.
there are only a few comments:
The authors should revise the introduction because in some places it is unclear.
Line 126 C. auris must be written in italics
Line 127 delete was
Author Response
Authors would like to cordially thank the reviewers for their valuable and helpful comments.
1. The authors should revise the introduction because in some places it is unclear
Response: We revised the introduction and improved the wording. All changes have been highlighted.
2. Line 126 C. auris must be written in italics
Response: C. auris is now written in italics.
3. Line 127 delete was
Response: “Was” was deleted
Reviewer 2 Report
The article deals with an important and relevant issue. Antifungal resistance of Candida auris strains due to their ability to form biofilm poses major challenges to the effective treatment of infections with these strains. In this context, testing the effectiveness of the products and methods used to disinfect surfaces contaminated with Candida biofilm as a measure to prevent the transmission of infection with this strain is extremely important.
The article as a whole is well structured, it is written in a clear and concise manner. However, some additional clarifications would be desirable.
For example, in Table 1, column 3, the concentration of the active ingredient, labeled 1, is considered to be the available measured chlorine concentration as specified below table. In the case of PAA 1 and 2 (peracetic acid), it can not be chlorine, so another parameter was probable measured. Please specify. How does NaDCC 1.2, 4 and 5 differ? They have the same concentration. If they are obtained from different commercial sources, the authors should specify this.
It would also be useful to specify the mechanism by which these disinfectants act.
Author Response
Authors would like to cordially thank the reviewers for their valuable and helpful comments.
1. The article as a whole is well structured, it is written in a clear and concise manner. However, some additional clarifications would be desirable. For example, in Table 1, column 3, the concentration of the active ingredient, labeled 1, is considered to be the available measured chlorine concentration as specified below table. In the case of PAA 1 and 2 (peracetic acid), it can not be chlorine, so another parameter was probable measured. Please specify.
Response: Peracetic acid concentration was measured by DPD method. We clarified that in the footer of Table 1.
2. How does NaDCC 1.2, 4 and 5 differ? They have the same concentration. If they are obtained from different commercial sources, the authors should specify this.
Response: We are not keen to actually name products as this is a research articles investigating the susceptibility of C. auris DSB. We have named the main ingredients and provided information of the different excipients when mentioned on label.
If the editor wants us to mention the name of the product we will oblige
3. It would also be useful to specify the mechanism by which these disinfectants act.
Response: The column with mechanism of disinfectant action was added into the Table 1 with corresponding references. 8 references were added.
Reviewer 3 Report
The authors tended to test the disinfection of some commercially available wipes and liquids against Candida auris, especially in the formation of dry biofilm. The effects of the tested samples were clearly reported. However, none of the mechanism was analyzed or discussed. Overall, this manuscript is more like a technical report rather than a research article.
Author Response
Authors would like to cordially thank the reviewers for their valuable and helpful comments.
1. The authors tended to test the disinfection of some commercially available wipes and liquids against Candida auris, especially in the formation of dry biofilm. The effects of the tested samples were clearly reported. However, none of the mechanism was analyzed or discussed. Overall, this manuscript is more like a technical report rather than a research article.
Response: The purpose of this article was 3-fold; i) explore the possibility to create reproducible C. auris DSB, ii) to investigate their susceptibility to currently use products and iii) investigate the impact of product use on dry biofilm transferability and eventually regrowth. The field of DSB is new and much remains to be done notably understanding the reasons why DSB are so resistant to disinfection and how products can eliminate DSB from surfaces while preventing transferred. We agree that mechanisms of action would be great to explore and no doubt this will be done in time. The column with mechanism of disinfectant action was added into the Table 1 with corresponding references. 8 references were added.
Reviewer 4 Report
p.p1 {margin: 0.0px 0.0px 0.0px 0.0px; font: 12.0px 'Helvetica Neue'} p.p2 {margin: 0.0px 0.0px 0.0px 0.0px; font: 12.0px 'Helvetica Neue'; min-height: 14.0px}
In this manuscript authors performed the testing of several disinfectants against biofilms of C. auris, an emerging pathogen.
This is a well thought work with clear results. My only comment is wondering if the authors performed SEM only to visualize the control biofilms? If so, What was the point of doing solely for that. It would be interesting to have some images of the several treatments performed.
Author Response
Authors would like to cordially thank the reviewers for their valuable and helpful comments.
1. This is a well thought work with clear results. My only comment is wondering if the authors performed SEM only to visualize the control biofilms? If so, What was the point of doing solely for that. It would be interesting to have some images of the several treatments performed.
Response: We provided the images of control biofilms to illustrate the reproducibility and structure of dry biofilm formed by sedimentation protocol. This is the first-time in vitro C. auris DSB are produced and visualised. We agree that presenting SEM images of the biofilm after disinfectant treatments would be very interesting and this will be done eventually.
Round 2
Reviewer 3 Report
The authors have imporved their writing.